# The Effects of High-Intensity Interval Training and Moderate Alcohol Consumption on Cognitive Performance—A Multidisciplinary Intervention in Young Healthy Adults

**DOI:** 10.3390/nu16111680

**Published:** 2024-05-29

**Authors:** Cristina Molina-Hidalgo, Francisco J. Amaro-Gahete, Jamie C. Peven, Kirk I. Erickson, Andres Catena, Manuel J. Castillo

**Affiliations:** 1Department of Medical Physiology, Faculty of Medicine, University of Granada, 18001 Granada, Spain; amarof@ugr.es (F.J.A.-G.); mcgarzon@ugr.es (M.J.C.); 2AdventHealth Research Institute, Neuroscience Institute, Orlando, FL 32804, USA; kirk.erickson@adventhealth.com; 3CIBER de Fisiopatología de la Obesidad y Nutrición (CIBEROBN), Instituto de Salud Carlos III, 28029 Granada, Spain; 4Instituto de Investigación Biosanitaria, Ibs.Granada, 18012 Granada, Spain; 5Behavioral Health Service Line, VA Pittsburgh Healthcare System, Pittsburgh, PA 15240, USA; jac349@pitt.edu; 6Department of Psychology, University of Pittsburgh, Pittsburgh, PA 15260, USA; 7Department of Experimental Psychology, Mind, Brain and Behavior Research Center (CIMCYC), University of Granada, 18011 Granada, Spain; acatena@ugr.es

**Keywords:** high-intensity training, diet, cognitive performance, young adults, alcohol consumption

## Abstract

Background. The main purpose of this study was to determine the effects of a high-intensity interval training (HIIT) intervention in the context of moderate alcohol consumption on cognitive performance in healthy young adults. Methods. We conducted a 10-week HIIT program along with four types of beverages with/without alcohol content. A total of 75 healthy adults (18–40 years old; 46% female) were allocated to either a control Non-Training group or an HIIT program group (2 days/week). Using block randomization, participants in the HIIT group were further allocated to an HIIT-Alcohol group (alcohol beer or sparkling water with vodka added, 5.4%) or an HIIT-NonAlcohol group (sparkling water or non-alcohol beer, 0.0%). The control group was instructed to maintain an active lifestyle but did not undergo any regular training. A comprehensive neuropsychological battery was used to evaluate cognitive performance (i.e., memory, working memory, processing speed, inhibitory control, and verbal fluency). Changes from baseline to week 10 were included in the main analyses. Results. All groups improved in all neuropsychological measures (all *p* ≤ 0.001), independent of sex and alcohol consumption, with no statistical differences between groups (all *p* > 0.05). Furthermore, larger increases in maximal oxygen uptake were associated with greater improvements in processing speed, inhibitory control, and verbal fluency (all *p* < 0.050). Conclusions. Although the improvements found in cognitive performance cannot be attributed to the HIIT intervention, no significant impairments in cognitive functions were noted due to moderate alcohol intake. Furthermore, our results confirmed that exercise-induced physical fitness improvements were associated with cognitive performance enhancements in young healthy adults.

## 1. Introduction

Exercise training is a powerful approach for improving cognitive function in both children and older adults [1,2,3]. In fact, it has been reported that cognitive function in the elderly is maintained or even improved following moderate-intensity exercise for at least 6 months [4]. Similarly, enhancements of visual–spatial and short-term memory have been reported after 6 to 12 weeks of programmed exercise in young adults [5,6]. A recent study has suggested that repeated high-intensity interval exercise (i.e., high-intensity interval training (HIIT)) could provide additional physiological and psychological adaptations in higher-order cognitive functions such as cognitive flexibility compared with lower-intensity exercise in older adults [7]. Similar improvements in cognitive flexibility have been found after a 7-week interval training intervention in young active individuals [8]. Complementary research indicates that an acute bout of interval exercise training can also positively impact cognitive function in healthy middle-aged individuals [9]. Nevertheless, there is still controversy regarding the optimal intensity of exercise training to improve cognitive performance in young adults [10]. One obvious reason for this absence of scientific literature is that cognitive health peaks during young adulthood, suggesting there is limited space for exercise-related cognitive improvement during this period [2].

Lifestyle habits, such as alcohol consumption, have been considered to play an important role in the development of dementia and other neurodegenerative diseases [11]. Alcohol intake, especially beer, is a common practice for many physically active people [12,13]. In addition to the deleterious effects of excessive consumption of alcohol on general health [11], there have been mixed results regarding the effects of low-to-moderate alcohol consumption on cognitive performance [14]. While Mehlig et al. suggested that low-to-moderate alcohol consumption increases the risk of developing cognitive impairments [15], other studies have found negligible effects associated with alcohol consumption [16,17]. Additionally, several studies have found significant improvements in cognitive function associated with low-to-moderate alcohol intake [18,19]. Due to the high heterogeneity of the above-mentioned results, investigating whether moderate alcohol consumption could influence the potential positive effects of exercise training on cognitive function is of scientific and clinical interest. To the best of our knowledge, no study has explored the combination of an HIIT intervention and moderate alcohol consumption in healthy individuals, which is common in a social context for active people.

The primary purpose of the present study was to investigate whether a highly demanding training intervention would improve cognitive function (i.e., memory, working memory, processing speed, inhibitory control, and verbal fluency) in healthy young adults, and whether those potential positive effects may be influenced by moderate alcohol consumption or by exercise-induced changes in other health-related parameters (cardiorespiratory fitness assessed by maximal oxygen uptake (VO_2_max), muscular strength determined by the handgrip test, and body composition). We predicted that concurrent regular alcohol intake, even in moderate amounts, could blunt any of the positive effects of training. Furthermore, we hypothesized that higher fitness levels would be associated with greater cognitive performance in young healthy adults.

## 2. Materials and Methods

### 2.1. Trial Design

The BEER-HIIT study is a registered controlled trial (ClinicalTrials.gov ID: NCT03660579) with a follow-up of 10 weeks. The procedures were designed in accordance with the last revised Declaration of Helsinki and approved by the Ethics Committee on Human Research of the University of Granada (321/CEIH/2017).

### 2.2. Participants

Eligible participants of the BEER-HIIT project were healthy young adults who lived in the province of Granada, Spain. The study was announced via social networks, local media, and posters. Prior to enrolment, all individuals provided written informed consent, completed a medical examination, and were fully informed about the study objectives, design, inclusion criteria, assessments to be undertaken, exercise program intervention, and types of beverages to be ingested. Subjects who met the inclusion criteria (i.e., (i) having a body mass index (BMI) from 18.5 to 30 kg/m^2^, (ii) not being engaged in a previous structured training program or a weight-loss program (in the last 5 months), (iii) having a stable body weight during the last 5 months (body weight changes < 3 kg), (iv) being free of disease, (v) not being pregnant or lactating, (vi) not taking any medication for chronic diseases, and (vii) not suffering pain, recent injuries, or other problems preventing strenuous physical activity) were invited to an information meeting in which the research staff gave specific information about healthy dietary patterns and the physical activity recommendations provided by the World Health Organization [20].

### 2.3. Randomization and Follow-Up

After the baseline measurements, a total of 83 individuals were allocated to a training (i.e., HIIT) or a non-training control group based on personal preferences. Participants in the control group were instructed to maintain their usual physical activity levels and not engage in a structured exercise program. Those participants included in the training group were subsequently allocated to an ethanol-containing beverage (i.e., 5.4% alcohol) group or an alcohol-free beverage group. The participants willing to consume ethanol were randomly allocated to either a group consuming alcohol beer (HIIT-Beer) or to a group consuming sparkling water with added vodka ethanol (HIIT-Alcohol). The participants choosing non-alcoholic beverages were randomly allocated to either an alcohol-free beer group (HIIT-0.0Beer) or a sparkling water group (HIIT-Water). This type of non-random (i.e., based on individual preference) and random allocation of the participants was conducted following ethical considerations and advice provided by the ethical committee (321-CEIH-2017), since drinking alcohol or participating in a highly demanding training program should be a personal choice.

### 2.4. Intervention

The HIIT intervention consisted of 2 sessions/week performed from Monday to Friday over 10 consecutive weeks with at least 48 h of recovery between sessions. The training intervention was divided into two different phases, starting with a familiarization phase to learn the main movement patterns, aiming to avoid injuries or potential dropouts. The volume and intensity of the sessions in the familiarization phase were fixed at 40 min/week and 8–9 Rating of Perceived Exertion (0–10 RPE), respectively [21,22]. Subsequent increments in both volume and intensity were established in Phase I (50 min/week and 10 RPE) and in Phase II (65 min/week and 10 RPE). Eight self-loading exercises were performed in a circuit form twice per set (i.e., frontal plank, high knees up, TRX horizontal row, squat, deadlift, side plank, push up, and burpees) with a passive rest between exercises and an active rest between sets (i.e., 6 RPE intensity, which corresponds to 60% VO_2_max) [22,23]. A dynamic standardized warm-up and an active global-stretching cooling-down protocol were completed at the beginning and the end of each training session, respectively. A detailed description of each exercise of the training program can be found elsewhere [24].

During the intervention, the alcohol consumption allowed was 330 mL of the respective beverage at lunch and 330 mL at dinner for men, and 330 mL at dinner for women, from Monday to Friday: (i) the HIIT-Alcohol group ingested randomly assigned alcohol beer (5.4% alcohol—Alhambra Especial^®^, Granada, Spain) or sparkling water with the exact equivalent amount of distilled alcohol added (vodka, 37.5% ethanol and 62.5% water), and (ii) the HIIT-NonAlcohol group was randomly assigned to ingest alcohol-free beer (0.0% alcohol—Cruzcampo^®^, Sevilla, Spain) or sparkling water (Eliqua 2^®^, Font Salem, Valencia, Spain). The amount of alcohol selected to ingest was based on scientific evidence (i.e., 2–3 drinks/day or 24–36 g of ethanol/day for men and 1–2 drinks/day or 12–24 g of ethanol/day for women) [25,26]. During weekends, participants were requested to respect the beverage intake condition (i.e., moderate alcohol consumption and non-alcohol consumption). All beverages were coded and provided by a blinded staff member of our research laboratory at the beginning of each week. Additionally, they were asked to report, before and after the 10-week intervention program, their usual frequency of alcohol intake through seven possible response categories using the Beverage Intake Questionnaire (BEVQ) [27,28]. Similarly, dietary habits were assessed using the MEDAS questionnaire [29].

### 2.5. Cognitive Function

Cognitive variables were taken at baseline and after 10 weeks of the supervised HIIT program. We used the Spanish Complutense Verbal Learning Test (TAVEC) to evaluate episodic memory, as well as its codification processes, data storage, and retrieval [30]. The main outcomes of this test were as follows: (i) learning process (defined as the sum of correctly recalled words across all five learning trials); (ii) short-term memory (free recall of list A after an interference list was presented); (iii) delay memory (free recall of list A after a 20 min rest period); and (iv) recognition (defined as total correct score (tc)) and discriminability index (calculated as the difference between correctly detected words from list A minus false alarms to new words (id)). Working memory was assessed using the Letter–Number Sequencing test, according to the instructions of the WAIS-IV manual [31], where the span score ranges from 0 to 7 and total scores range from 0 to 21. We measured processing speed and inhibitory control using the one-page paper-and-pencil cancelation test version of the D2 test [32]. In addition, we used a standard verbal fluency test to measure phonemic, semantic, and total verbal fluency [33]. Higher scores indicate better cognitive performance on these tasks. Comprehensive information on the cognitive measurements can be found in the study protocol [24].

Age, sex, and occupational activity were also registered by a self-report demographic questionnaire before the intervention.

### 2.6. Statistical Analyses

These results presented in the current study were not the main outcomes of the BEER-HIIT project [34,35].

Our sample size calculations revealed that 13 participants per group were needed to detect an effect size of 0.25 in memory scores with an α error of 0.05 and a power of 0.85 [36]. However, a minimum of 16 participants per group (a total of 80) were recruited, allowing us to consider a maximum loss of 20% at the follow-up [37].

Standard statistical methods were used for the calculation of means and standard deviations. Normal Gaussian distribution of the data was verified by the Shapiro–Wilk test and visual check of histograms, Q–Q, and box plots. Homoscedasticity was verified by the modified Levene test. The compound symmetry, or sphericity, was checked by the Mauchly test. When the assumption of sphericity was not met, the significance of F-ratios was adjusted according to the Greenhouse–Geisser procedure when the epsilon correction factor was <0.75, or according to the Huyn–Feld procedure when the epsilon correction factor was >0.75. 

Composite scores for memory (immediate, short-term, delayed recall, and recognition scores; α = 0.900), working memory (direct and processing scores; α = 0.922), processing speed (total productivity, correct work, and concentration index scores; α = 0.986), inhibitory control (total effectiveness score), and verbal fluency (total score) were calculated by averaging z scores for their individual components. Recategorization of the occupational activity measure was performed into three levels, showing a relatively high internal consistency (α = 0.705). 

Age, sex, and occupational activity were used as potential confounders in the analyses. We combined the responses of computed and classified occupational activities and categorized them as level 1 (unemployment, homemaker, and student), level 2 (primary sector services, retail or catering services, and administrative clerk), and level 3 (support and scientific/intellectual technicians and professionals, and business administration and management). Given that we did not observe a sex interaction, we conducted the analysis including males and females together. Similarly, since no beverage interaction was observed between groups in any outcome, we analyzed both alcohol beer and sparkling water with ethanol HIIT groups in the same group (i.e., HIIT-Alcohol group) and both 0.0% alcohol beer and sparkling water HIIT groups in the same group (i.e., HIIT-NonAlcohol group).

A repeated-measures analysis of variance (2 × 2 ANOVA; time × training group; and time × beverage) tested group differences over time. Post hoc Bonferroni corrections were conducted for multiple comparisons. The significance level was set at *p* < 0.05 for all analyses. Finally, simple and multiple linear regression analyses were performed to examine the associations between changes in physical fitness (i.e., maximum oxygen uptake, hand grip strength, and body composition, such as BMI, FMI, and LMI) with changes in cognitive outcomes (i.e., memory, working memory, processing speed, inhibitory control, and verbal fluency) after the intervention. Age, sex, and occupational activity were included as potential confounders, which were selected based on statistical procedures (i.e., hierarchical regressions) and theoretical bases. 

All analyses were conducted using the Statistical Package for Social Sciences (SPSS, v. 25.0, IBM SPSS Statistics, IBM Corporation, Armonk, NY, USA). Graphical presentations were prepared using GraphPad Prism 8 (GraphPad Software, San Diego, CA, USA).

## 3. Results

A total of 74 participants (34 women) were included in the final analyses after a loss to follow-up of 10% (see Figure 1). The characteristics of the participants are presented in Table 1. There were no significant differences between groups at baseline (all *p* > 0.05), except for age and occupational activity (*p* = 0.006 and *p* = 0.005, respectively; see Table 1). Our HIIT intervention has previously been shown to successfully improve physical fitness parameters, including absolute and relative VO_2_max [35], as well as body composition parameters [34].

### 3.1. Intervention Effects of an HIIT Program on Cognitive Function

Figure 2 shows changes in learning (number of recalled words across all five trials) tested at baseline and post-intervention. A significant increase in the number of recalled words across trials at baseline and post-intervention measures was observed independently of the intervention group (all *p* ≤ 0.001; see Figure 2). Significant differences in learning were also noted between baseline and post-intervention measures in the Non-Training group, HIIT-Alcohol group, and HIIT-NonAlcohol group (all *p* ≤ 0.001; see Figure 2). 

Table 2 presents changes in the raw scores and z-transformed cognitive outcomes before and after the intervention. The z-scores are interpreted as the change from baseline in standard deviations. The Non-Training, HIIT-Alcohol, and HIIT-NonAlcohol groups all showed significant improvements in memory (i.e., immediate, short-term, and delayed recall scores) and recognition (all *p* ≤ 0.001; see Table 2) after the intervention. Similarly, all groups showed significant enhancements in processing speed, inhibitory control, and verbal fluency (all *p* ≤ 0.01; see Table 2). There were no within-group differences in memory, working memory, processing speed, inhibitory control, or verbal fluency z-scores (all *p* > 0.05; see Table 2). The ANCOVA of raw scores and z-transformed cognitive outcomes, adjusting for baseline values (Model 0), showed no significant between-group differences in any cognitive function outcomes (all *p* > 0.05; see Table 2). These results persisted after controlling for potential confounders (i.e., age, sex, and occupational activity; see Table A1). 

### 3.2. Are Exercise-Induced Changes in Cognitive Function Explained by Those Obtained in Physical Fitness and Body Composition after the Intervention?

Higher increases in relative VO_2_max (mL/kg/min) over the course of the intervention were associated with greater improvements in processing speed, inhibitory control, and verbal fluency (all β > 0.249, all *p* values < 0.034). These statistically significant relationships persisted after adjusting for sex, age, and occupational activity (see Figure 3). Finally, changes in verbal fluency were positively related to changes in absolute VO_2_max (mL/min) (β = 0.313, R^2^ = 0.085, *p* = 0.007; see Table A3), which remained after adjusting for sex, age, and occupational activity (all *p* ≤ 0.013; see Table A3). Additionally, higher BMI levels were associated with poorer working memory (composite score) (β = −0.282, R^2^ = 0.066, *p* = 0.016; see Figure 3D), which also persisted after adjusting for sex, age, and occupational activity (all *p* ≤ 0.019; see Figure 3D). No significant associations were observed between changes in cognitive outcomes and changes in hand grip strength, fat mass index, or lean mass index (see Table A2 and Table A3).

## 4. Discussion

The main aim of this study was to investigate whether a high-intensity training intervention would improve cognitive function (i.e., memory, working memory, processing speed, inhibitory control, and verbal fluency) and whether those potential positive effects would be influenced by regular moderate alcohol consumption in healthy young adults. Additionally, we examined whether improvements in cognitive performance were related to other positive health-related parameters (e.g., VO_2_max and body composition). Our results did not completely support the hypothesis that the HIIT intervention improved learning, memory, processing speed, inhibitory control, and verbal fluency, since the Non-Training group exhibited similar enhancements in these parameters. However, interestingly, our findings suggest that regular moderate alcohol consumption for this duration does not negatively influence cognitive function. Additionally, in line with our hypothesis, the current findings showed that physical fitness improvements after our 10-week HIIT intervention were associated with better cognitive performance in young healthy adults.

### 4.1. Intervention Effects of an HIIT Program on Cognitive Function

Previous research has demonstrated that there is moderate evidence supporting the notion that physical activity, in the form of aerobic exercise, benefits cognitive functioning during early and late periods of life [1,3,38]. Regarding a high-intensity exercise program, prior studies have found that short-term HIIT resulted in meaningful improvements in reaction time and cognitive flexibility in older adults [7], and in learning performance in healthy, active male students [39]. These enhancements have been attributed to physiological processes that occur during exercise, including an increase in catecholamines (i.e., dopamine, epinephrine, and norepinephrine) and release of neurotrophic factors [39]. Although these results are encouraging, there are still major gaps in our understanding of the effects of an HIIT intervention on cognitive performance in young to middle-aged adults (aged 18–50 years) [1,3]. In this regard, although cognitive improvements were found in all cognitive domains measured (i.e., learning, memory, processing speed, inhibitory control, and verbal fluency), we cannot conclude that those benefits were a consequence of our HIIT intervention since the Non-Training control group showed similar performance improvements. These findings are somewhat consistent with those obtained by similar investigations applying HIIT interventions that reported no exercise-induced changes in working memory after 7–12 weeks of an HIIT program in healthy middle-aged individuals [8,9,40]. Hence, an important question remains unanswered regarding the inconsistent findings in cognitive performance improvements following exercise interventions among young and middle-aged adults. One reason for this heterogeneity could be that cognitive health peaks during young adulthood, which would suggest a ceiling effect for exercise-related improvements to cognitive function during this period of life [2]. Other reasons that should be considered may be variability in methodological and study design factors, such as (i) the general health and fitness level of participants being enrolled, (ii) the neuropsychological tests used to measure aspects of cognition, (iii) the lack of consistent reporting of blinding and adherence/compliance, or (iv) the nature of control groups. Indeed, many of the instruments used to assess executive functioning are traditional neuropsychological tools primarily developed to aid in clinical diagnosis rather than to assess individual variation in normative cognitive functioning; thus, their sensitivity to detect changes in response to an intervention (especially in the context of a normative sample) remains questionable [1].

In addition to its role in physical health, low-to-moderate alcohol consumption has been suggested as a potential risk factor in the development of cognitive impairment, which is also highly associated with cardiovascular diseases [18]. However, some inconsistencies can be observed across studies. Zhang et al. (2020) [18] hypothesized that the impact of alcohol drinking on cognitive function may be dependent on a balance of its beneficial and harmful effects on the cardiovascular system [18]. However, other studies have reported that moderate drinkers are more likely to have hippocampal atrophy [41] suggesting that, in line with other studies [14], there is no safe level of drinking when referring to health. These variable results could arise from the complicated and plural mechanisms through which alcohol consumption affects health [14], as well as the influence of an individual’s consumption volume and pattern of drinking [42]. Our findings suggest that moderate alcohol consumption did not impair cognitive function after 10 weeks in healthy young adults who engaged in vigorous exercise training.

In summary, there is still controversy regarding the effects of moderate-to-vigorous intensity exercise on cognition in young and middle-aged adults. Therefore, future studies are needed to better elucidate the potential benefits of exercise on cognition during young and middle adulthood [1]. Furthermore, the association of low-to-moderate drinking with cognitive function in the younger ages, as well as the mechanisms underlying this potential association, warrants additional investigation.

### 4.2. Are Exercise-Induced Changes in Cognitive Function Explained by Those Obtained in Physical Fitness and Body Composition after the Intervention?

Previous studies have confirmed that an enhancement of cardiovascular fitness is associated with changes in cognitive performance in children and older adults [1,2,3,43,44]. However, it is unclear precisely what function cardiovascular fitness might play in instantiating these cognitive changes [45]. Regarding the effects of physical activity on cognition and brain outcomes in young middle-aged adults (18–59 years old), there is a dearth of high-quality data available. Our results partially align with previously demonstrated evidence of a positive relationship between changes in cardiorespiratory fitness and improvements in cognitive functions (e.g., processing speed, inhibitory control, and verbal fluency) being found in a healthy young adult cohort. Although our effect sizes were relatively small, these findings suggest that cardiorespiratory fitness may impact challenging and complex cognitive processes, which has previously been demonstrated in other samples and age ranges [46]. However, future research is needed to bring to light whether exercise training (i.e., HIIT) can optimize cognitive function in young and middle-aged adults.

We found a negative association between BMI levels and working memory, which is consistent with some previously reported evidence from childhood [47] to adulthood [48]. Although specific cognitive deficits have been demonstrated in younger adults and older adults, the findings are inconsistent across age groups. Overall, the results found in the current study could extend our understanding of the possible neuropathological processes underlying obesity-related cognitive dysfunction, demonstrating a specific impact on predominant prefrontal cognitive processes.

In summary, our study has demonstrated a notable advantage to implementing a supervised high-intensity exercise intervention, as evidenced by an 80% adherence rate among participants, which resulted in significant improvements in both physical fitness and body composition [34,35].

### 4.3. Limitations

Although this study had several strengths, it also had several limitations that should be considered. Firstly, the control group was not purely sedentary since participants were instructed to maintain an active lifestyle. Although physical activity levels were monitored, they were determined using self-reported approaches. Objective methods (i.e., accelerometer systems) might result in different outcomes. Second, the BEER-HIIT project was designed from a quantitative approach that looks forward to modifying physical fitness and/or additional health-related outcomes. Thus, changes in cognitive functions are only expected as a result of physical fitness enhancements [10]. The third limitation was our sample size. Finally, participants were not fully randomized because of ethical considerations regarding alcohol consumption. Thus, a truly double-blind design, placebo-controlled for alcohol, was not possible. Subsequently, future studies are needed to determine the effects of the same training intervention on participants with different biological characteristics, using reliable, objective measurements to identify effective public health strategies that promote a healthy lifestyle where fermented beverages in moderate amounts co-ingested with meals could be included if wished and considered acceptable.

## 5. Conclusions

Although our 10-week HIIT exercise intervention failed to moderate the impact on cognitive performance, moderate alcohol intake at this level and for this duration did not seem to alter cognitive performance in young healthy adults. Furthermore, our results confirmed that exercise-induced changes in physical fitness were associated with improvements in cognitive performance in young healthy adults.

## Figures and Tables

**Figure 1 nutrients-16-01680-f001:**
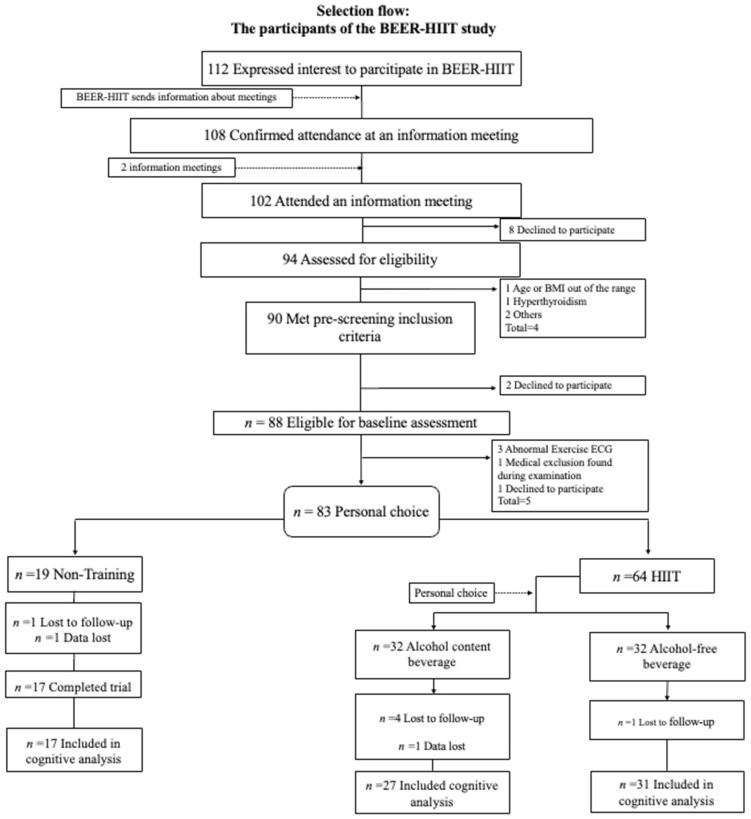
Flow of participants in the BEER-HIIT study. Abbreviations: HIIT, high-intensity interval training; BMI, body mass index; ECG, electrocardiogram.

**Figure 2 nutrients-16-01680-f002:**
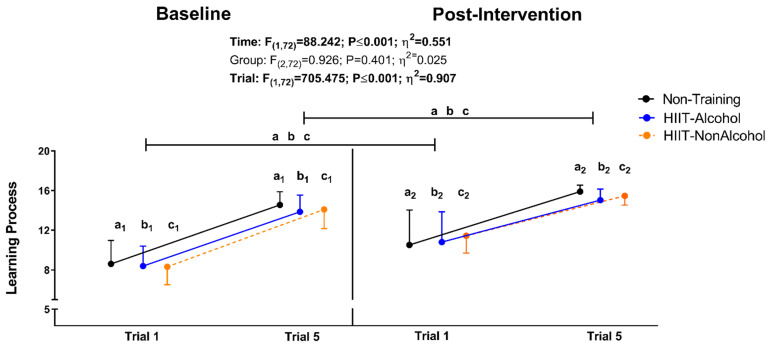
Changes in learning process (Trial 1 and 5) were measured by the Spanish Complutense Verbal Learning Test (TAVEC), tested at baseline and post study intervention. F, *p*, and η of repeated-measures analysis of variance (ANOVA) for time, group, and trial. Intragroup changes obtained by repeated-measures ANOVA (group × time × trial, adjusting by trial), with a post hoc Bonferroni-corrected test, are indicated as Trial 1 vs. Trial 5 (a_1_ and a_2_; all *p* < 0.05) for the Non-Training group; Trial 1 vs. Trial 5 (b_1_ and b_2_; all *p* < 0.05) for the HIIT-Alcohol group; the Trial 1 vs. Trial 5 (c_1_ and c_2_; all *p* < 0.05) for the HIIT-NonAlcohol group. Intragroup trials’ changes before and post-intervention obtained by repeated-measures ANOVA (group × time × trial, adjusting by time), with a post hoc Bonferroni-corrected test, are indicated as *p* < 0.05 for the Non-Training group, b *p* < 0.05 for the HIIT-Alcohol group, and c *p* < 0.05 for the HIIT-NonAlcohol group. Raw data (total correct responses) are presented as the mean and standard deviation. Abbreviations: HIIT-Alcohol, group that performed high-intensity interval training and consumed alcoholic beverages; HIIT-NonAlcohol, group that performed high-intensity interval training and consumed non-alcoholic beverages.

**Figure 3 nutrients-16-01680-f003:**
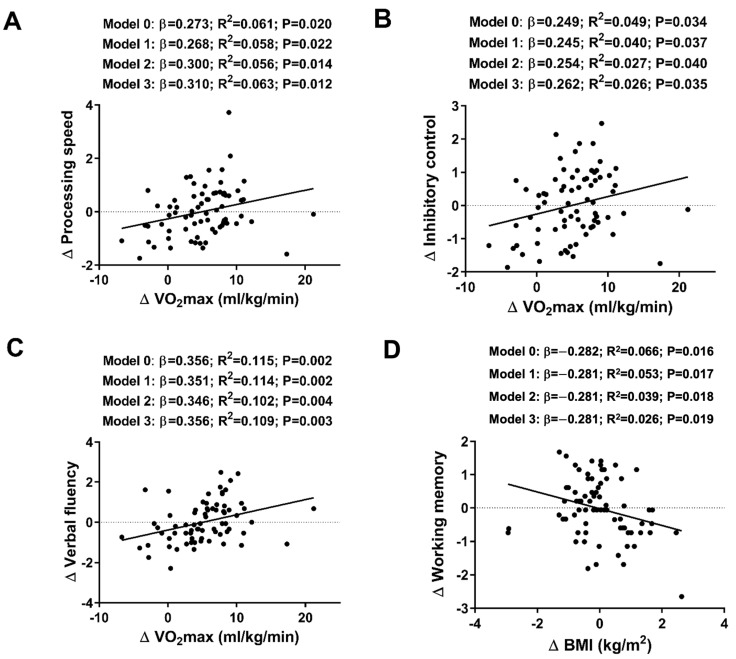
Associations between maximal oxygen uptake (VO_2_max) (**A**) and processing speed (composite score); VO_2_max (**B**) and inhibitory control (z-score); VO_2_max (**C**) and verbal fluency (total z-score), and body mass index (BMI) adjusted for age, sex, and occupational activity (**D**) and working memory (composite score) in young healthy adults. The analyses were controlled for: age (Model 1); both age and sex (Model 2); and age, sex, and occupational activity (Model 3). The β (standardized linear regression coefficient), R^2^ (coefficient of determination), and *p* value were obtained from the linear regression analyses.

**Table 1 nutrients-16-01680-t001:** Baseline characteristics of participants.

	Non-Training (*n* = 17)			HIIT-Alcohol (*n* = 27)			HIIT-NonAlcohol (*n* = 31)			
	Mean ± SD									*p* Value
Sex (men%/women%)	64.7%/35.3%			48.1%/51.8%			54.8%/45.2%			0.572
Age	20.2 ± 1.3			24.2 ± 1.5			24.5 ± 1.5			**0.006 ***
Alcohol Ingested (mL/week)	1144.1 ± 831.9			696.7 ± 857.4			1204.5 ± 1167.3			0.131
Educational level (%)										0.097
Primary Education	--			3.7%			--			
Secondary Education	47.1%			40.7%			22.5%			
Vocational Education and Training	23.5%			18.5%			22.5%			
University Degree or Certificate of Higher Education	29.5%			37.3%			54.8%			
Occupational activity (%)										**0.005 ***
Level 1		100% †			55.6%			54.8%		
Level 2		--			18.5%			16.1%		
Level 3		--			25.9%			29.1%		
Learning Process										
Trial 1	8.8 ± 2.3			8.3 ± 2.0			8.3 ± 1.8			0.666
Trial 2	11.2 ± 2.7			12.2 ± 2.1			11.4 ± 2.3			0.269
Trial 3	12.9 ± 2.3			13.1 ± 2.2			12.4 ± 2.7			0.530
Trial 4	14.4 ± 1.3			13.4 ± 2.1			13.4 ± 3.1			0.329
Trail 5	14.5 ± 1.3			13.8 ± 1.7			14.1 ± 1.9			0.474
Memory										
Immediate memory	61.8 ± 7.4			60.9 ± 7.7			59.6 ± 9.3			0.638
Short-term memory	13.6 ± 1.2			13.9 ± 1.8			12.9 ± 2.5			0.170
Delay memory	13.9 ± 1.4			14.2 ± 1.8			13.3 ± 2.2			0.224
Recognition (tc)	15.3 ± 0.9			15.6 ± 0.6			15.0 ± 1.4			0.115
Recognition (id)	97.7 ± 2.7			98.2 ± 1.9			96.9 ± 4.1			0.284
Working Memory										
Direct Score	13.0 ± 1.6			11.8 ± 2.1			11.7 ± 2.5			0.125
Processing Score	5.8 ± 0.8			5.6 ± 0.9			5.5 ± 0.9			0.507
Processing Speed										
Total productivity	516.6 ± 88.0			514.5 ± 76.5			510.9 ± 67.3			0.965
Correct work	208.7 ± 43.4			203.9 ± 39.0			200.7 ± 36.5			0.793
Concentration Index	208.4 ± 43.5			203.4 ± 39.4			200.0 ± 36.6			0.776
Inhibitory Control										
Total Effectiveness	502.9 ± 86.7			498.3 ± 77.5			492.7 ± 68.1			0.900
Verbal Fluency										
Phonologic	86.1 ± 23.2			88.4 ± 20.5			77.1 ± 15.0			0.069
Semantic	40.5 ± 8.0			40.1 ± 6.5			37.5 ± 8.5			0.315
Total Score	126.6 ± 28.0			128.4 ± 23.8			114.6 ± 19.3			0.057

Data expressed the as mean ± standard deviation. Differences in baseline characteristics between the different groups were evaluated with analysis of variance (ANOVA). † Occupational activity: 53% unemployment, 47% student. * Boldface values indicate significant differences (*p* < 0.05). Abbreviations: HIIT-Alcohol group that performed high-intensity interval training and consumed alcoholic beverages; HIIT-NonAlcohol group that performed high-intensity interval training and consumed non-alcoholic beverages; Level 1, unemployment, homemaker, and student; Level 2, agriculture, livestock and fishery services, retail or catering services, and administrative clerk; Level 3, support technicians and professionals, scientific and intellectual technicians and professionals, business administration, and management.

**Table 2 nutrients-16-01680-t002:** Changes before and after the intervention study and changes over time in cognitive scores.

	Non-Training			HIIT-Alcohol			HIIT-NonAlcohol					
	Baseline	Post-Intervention		Baseline	Post-Intervention		Baseline	Post-Intervention		Model 0		
	Mean ± SD											
Memory			*p* Value			*p* Value			*p* Value	F	*p* Value	η^2^
Immediate Memory	61.82 ± 7.38	70.06 ± 5.85	**≤0.001**	60.93 ± 7.68	70.15 ± 5.56	**≤0.001**	59.55 ± 9.26	67.94 ± 6.87	**≤0.001**	0.788	0.459	0.022
Short-Term Memory	13.59 ± 1.23	15.29 ± 0.85	**≤0.001**	13.85 ± 1.81	14.89 ± 1.58	**0.004**	12.87 ± 2.47	14.77 ± 1.45	**≤0.001**	0.850	0.432	0.023
Delay Memory	13.88 ± 1.41	15.47 ± 0.62	**≤0.001**	14.15 ± 1.83	14.93 ± 1.62	**0.0014**	13.29 ± 2.18	15.00 ± 1.13	**≤0.001**	2.043	0.137	0.054
Recognition (tc)	15.29 ± 0.92	16.00 ± 0.00	**0.005**	15.56 ± 0.64	15.89 ± 0.32	0.105	14.97 ± 1.38	15.68 ± 0.70	**≤0.001**	1.956	0.149	0.052
Recognition (id)	97.73 ± 2.67	99.60 ± 0.89	**0.014**	98.23 ± 1.93	98.82 ± 1.93	0.321	96.92 ± 4.09	99.12 ± 1.73	**≤0.001**	1.632	0.203	0.044
Composite score	0.082 ± 0.204	0.277 ± 0.174	0.315	0.195 ± 0.162	−0.002 ± 0.162	0.201	−0.219 ± 0.151	−0.125 ± 0.129	0.513	1.462	0.239	0.040
Working Memory												
Direct Score	13.00 ± 1.62	13.06 ± 3.47	0.928	11.78 ± 2.08	11.96 ± 3.11	0.722	11.74 ± 2.49	11.77 ± 2.97	0.947	0.116	0.891	0.003
Processing Score	5.82 ± 0.81	5.88 ± 1.27	0.845	5.63 ± 0.88	5.41 ± 1.12	0.354	5.52 ± 0.89	5.32 ± 1.49	0.387	0.574	0.566	0.016
Composite score	0.349 ± 0.820	0.307 ± 1.020	0.852	−0.042 ± 0.186	−0.050 ± 0.183	0.966	−0.117 ± 0.173	−0.113 ± 0.171	0.982	0.187	0.830	0.005
Processing Speed												
Total productivity	516.59 ± 87.93	576.59 ± 72.84	**≤0.001**	514.52 ± 76.49	566.07 ± 67.83	**≤0.001**	510.87 ± 67.316	571.87 ± 52.41	**≤0.001**	0.691	0.504	0.019
Correct work	208.71 ± 43.44	244.82 ± 39.72	**≤0.001**	203.85 ± 39.04	235.15 ± 36.43	**≤0.001**	200.68 ± 36.54	237.29 ± 39.76	**≤0.001**	0.530	0.591	0.015
Concentration Index	208.41 ± 43.46	242.94 ± 41.62	**≤0.001**	203.44 ± 39.37	234.85 ± 36.50	**≤0.001**	200.00 ± 36.58	236.94 ± 39.72	**≤0.001**	0.442	0.644	0.012
Composite score	0.102 ± 0.245	0.136 ± 0.240	0.783	0.007 ± 0.194	−0.075 ± 0.180	0.399	−0.067 ± 0.181	−0.007 ± 0.178	0.516	0.618	0.542	0.017
Inhibitory Control												
Total Effectiveness	502.82 ± 86.61	565.18 ± 74.533	**≤0.001**	498.26 ± 77.45	552.22 ± 66.00	**≤0.001**	492.61 ± 68.01	554.87 ± 55.62	**≤0.001**	0.713	0.494	0.020
Verbal Fluency												
Phonologic	86.12 ± 23.15	95.71 ± 24.51	**0.003**	88.37 ± 20.55	96.89 ± 16.70	**≤0.001**	77.10 ± 14.93	84.42 ± 17.16	**0.002**	0.964	0.386	0.026
Semantic	40.47 ± 8.02	39.88 ± 10.78	0.705	40.11 ± 6.47	42.37 ± 6.03	0.070	37.48 ± 8.55	37.48 ± 6.74	1.000	2.364	0.101	0.062
Total Verbal Fluency	126.59 ± 27.97	135.59 ± 33.55	**0.016**	128.41 ± 23.76	139.26 ± 21.09	**≤0.001**	114.58 ± 19.25	121.90 ± 20.06	**0.008**	1.103	0.338	0.030

*p* value of intragroup changes before and post-intervention obtained by repeated-measures ANOVA. F, *p* value, and η^2^ of analysis of covariance adjusting by baseline values. Boldface values indicate significance differences (*p* < 0.05). Abbreviations: HIIT-Alcohol, group that performed high-intensity interval training and consumed alcoholic beverages; HIIT-NonAlcohol, group that performed high-intensity interval training and consumed non-alcoholic beverages; tc, total number of correct responses in the recognition test; id, index of discriminability from learning list in the recognition test.

## Data Availability

The raw data supporting the conclusions of this article will be made available by the authors on request.

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
