# Peer review of "The Effects of High-Intensity Interval Training and Moderate Alcohol Consumption on Cognitive Performance—A Multidisciplinary Intervention in Young Healthy Adults"

_nutrients, 2024, doi:10.3390/nu16111680_

Round 1

Reviewer 1 Report

Comments and Suggestions for Authors

The title could be without a question mark. It is better to highlight the research problem than the doubts.

The introduction is enough. The primary goal of this study is correct. What about exercise-induced changes in other health-related parameters? Did the authors forget about this goal in the results? Where are these results?

Do the criteria include alcohol consumption/experience? What happened before the experiment started?

I would like to know how many test trials were performed before the main measurement to obtain the baseline results. Was there familiarization with each test? How many times? This may affect the increase results from the first to the last measurement.

Sample size calculations indicated that 13 participants per group were needed to detect an effect size of 0.25. This is a very small effect. Could you interpret the significance of this effect for science? There are no units in the table and figures. It's hard to keep up. The conclusions are inconclusive because the quality of the measurements is questionable. The learning processes! But the authors can improve the description of the methods and it can make a difference.

Author Response

Point-by-point response to Comments and Suggestions from Reviewer 1:

Comment 1: The title could be without a question mark. It is better to highlight the research problem than the doubts.

Response 1: Thank you for pointing this out. We agree with this comment. Therefore, we have removed the question mark from the title. “[The effects of high-intensity interval training and moderate alcohol consumption on cognitive performance - A multidisciplinary intervention in young healthy adults]”

Comment 2: The introduction is enough. The primary goal of this study is correct. What about exercise-induced changes in other health-related parameters? Did the authors forget about this goal in the results? Where are these results?

Response 2: Thank you for pointing this out. The exercise-induced changes in other health-related markers included in the current study have been previously published. We have added a few sentences to summary these results in the main text. “[Our HIIT intervention has previously been shown to successfully improve physical fitness parameters, including absolute and relative VO2max,[35] as well as body composition parameters.[34]]”

Comment 3: Do the criteria include alcohol consumption/experience? What happened before the experiment started? 

Response 3: Thank you for this comment. We asked the participants to report, before and after the 10-week intervention program, their usual frequency of alcohol intake through seven possible response categories using the Beverage Intake Questionnaire (BEVQ). No differences among groups were found in alcohol consumption intake previous to the intervention program. This information is included in the main text [Ln 146-149].

Comment 4: I would like to know how many test trials were performed before the main measurement to obtain the baseline results. Was there familiarization with each test? How many times? This may affect the increase results from the first to the last measurement.

Response 4: Thank you for this comment. We did not perform any test trials prior to the main baseline measurement to avoid practice or ceiling effects. All the participants were measured twice, once at baseline and once 10 weeks after the exercise intervention. All analyses were adjusted by baseline levels to test the changes over time.

Comment 5: Sample size calculations indicated that 13 participants per group were needed to detect an effect size of 0.25. This is a very small effect. Could you interpret the significance of this effect for science? There are no units in the table and figures. It's hard to keep up. The conclusions are inconclusive because the quality of the measurements is questionable. The learning processes! But the authors can improve the description of the methods and it can make a difference.

Response 5: Thank you for this comment. We agree with the reviewer’s comment and understand the complexity of the results presented. In Figure 2 (Changes in learning process), we specified the units presented in the caption of the figure [Ln 249, “Raw data (# correct responses) are presented as mean and standard deviation]. In Table 2, we reported “F, p value, and η2 of analysis of covariance adjusting by baseline values”. The η2 values indicate the effect sizes for our groups over time comparisons. Since we did not find significant differences between groups over time, we did not discuss the small effect sizes obtained (all η2 < 0.06). Finally, we included R2 in our Figure 3, which represents the association between changes in fitness and body composition and cognitive performance, indicating the percentage of the variance in the dependent variable that the independent variable explains collectively. We indicated in the main text that our effect sizes are relatively small (all R2 < 0.115); thus, our results should be interpreted with caution [Ln 366-367].

Reviewer 2 Report

Comments and Suggestions for Authors

The study aims to investigate the effects of a 10-week High-Intensity Interval Training (HIIT) program on cognitive performance in healthy young adults, considering the context of moderate alcohol consumption. The introduction sets the scene but I would like the authors to say why exercise might have these positive effects. I would also like the authors to make a much stronger connection between cognitive performance and the measures used in their study. They mention Alzheimer’s but this is not realistic for the present study.  

I like the use of a randomized controlled trial design, which is appropriate for assessing intervention effects.

Alcohol Consumption Measurement: The study categorized alcohol consumption as moderate, but the definition and quantification of moderate intake could be clarified. I suspect for some people, the amount of alcohol was a great deal and for others, not much at all.

Cognitive performance. More details are needed on this assessment. Also, potential confounding factors  such as socioeconomic status, dietary habits should be considered and controlled for to isolate the effects of the intervention.

Alcohol Impact: While the study found no significant impairments in cognitive function due to moderate alcohol consumption, the study was quite short in duration and so may not capture long-term effects or variations in individual responses to alcohol. Or the alcohol consumption amount was not large enough. Justification that the amount of alcohol drunk is needed. Additionally, alcohol timings – please comment on these as surely alcohol is consumed more at the end of the day rather than in the middle?

The authors should use the future research aspect to explain a more rigorous test of the main hypotheses.

Comments on the Quality of English Language

It is well written. 

Author Response

Point-by-point response to Comments and Suggestions from Reviewer 2:

Comment 1: The study aims to investigate the effects of a 10-week High-Intensity Interval Training (HIIT) program on cognitive performance in healthy young adults, considering the context of moderate alcohol consumption. The introduction sets the scene but I would like the authors to say why exercise might have these positive effects. I would also like the authors to make a much stronger connection between cognitive performance and the measures used in their study. They mention Alzheimer’s but this is not realistic for the present study. 

Response 1: Thank you for this comment. Exercise has been shown to positively impact cognitive function in various settings and populations. Among several methodological considerations, exercise intensity has been highlighted as a crucial factor for these beneficial cognitive effects. For that reason, we hypothesized that our high-intensity interval intervention would positively affect cognitive function in our sample. However, there are still major gaps in our understanding of the possible mechanisms by which exercise exerts these positive effects on cognition. In our introduction and discussion sections, we tried to highlight these two main points. Additionally, we discussed one possible mechanistic pathway for these exercise-induced cognitive improvements, which could be enhancements in cardiovascular fitness.

Finally, we ensured that there is nothing related to Alzheimer's or dementia since, as the author mentioned, this does not apply to our young, healthy sample.

Comment 2: I like using a randomized controlled trial design appropriate for assessing intervention effects.

Response 2: Thank you for this comment. We did our best to create a study design that aligns with the suggestions of the Ethical Committee, despite the complexity of this type of intervention.

Comment 3: Alcohol Consumption Measurement: The study categorized alcohol consumption as moderate, but the definition and quantification of moderate intake could be clarified. I suspect for some people, the amount of alcohol was a great deal and for others, not much at all.

Response 3: Thank you for this comment. We based on scientific evidence to select the amount of alcohol ingested by the participants, which defines a moderate amount as two or three drinks/day or 24–36 g of ethanol/day for men (660 mL/day) and one to two drinks/day or 12–24 g of ethanol/day for women (330 mL/day)(Poli, A.; Marangoni, F.; Avogaro, A.; Barba, G.; Bellentani, S.; Bucci, M.; Cambieri, R.; Catapano, A.L.; Costanzo, S.; Cricelli, C.; et al. Moderate alcohol use and health: A consensus document. Nutr. Metab. Cardiovasc. Dis. 2013, 23, 487–504; Meister, K.A.; Whelan, E.M.; Kava, R. The Health Effects of Moderate Alcohol Intake in Humans: An Epidemiologic Review. Crit. Rev. Clin. Lab. Sci. 2000, 37, 261–296). We highlighted the following information in the main text [Ln 141-143].

Comment 4: Cognitive performance. More details are needed on this assessment. Also, potential confounding factors such as socioeconomic status, dietary habits should be considered and controlled for to isolate the effects of the intervention.

Response 4: Thank you for this comment. We attempted to condense the cognitive measurement details to fit within the journal's word limit. However, comprehensive information on the cognitive measurements can be found in the study protocol (24.   Molina-Hidalgo C., De-la-O A., Jurado-Fasoli, L Amaro-Gahete, F.J., Catena A., and Castillo.M. Investigating the Alcohol Effects on the Response to Strenuous Exercise Training: The BEER-HIIT Study. Substance Abuse & Addiction Journal 2024). We have added a reference to the study protocol for clarification [Ln 166].

We agree with the reviewer that these outcomes are significant and should be taken into consideration when testing cognitive outcomes. However, our small sample size limits the possibility of including more confounding factors in our models. We included the main confounders that are likely to affect our primary outcomes, such as age, sex, and occupational activity.

Comment 5: Alcohol Impact: While the study found no significant impairments in cognitive function due to moderate alcohol consumption, the study was relatively short in duration and so may not capture long-term effects or variations in individual responses to alcohol. Or the alcohol consumption amount was not large enough. Justification that the amount of alcohol drunk is needed. Additionally, alcohol timings – please comment on these as surely alcohol is consumed more at the end of the day rather than in the middle?

Response 5: Thank you for this comment. We completely agree with the reviewer in the short duration of our study to test the possible alcohol impairments on cognitive performance, and we ensured that this was reflected in our discussion and limitation section. We tried to highlight the inconsistencies found in previous studies due to the different alcohol amounts ingested, individual responses to alcohol due to physiological conditions, or previous alcohol consumption history. Regarding the time that the alcohol is consumed, we designed a protocol to standardize the consumption timing, simulating a typical moderate consumption in the context of the Mediterranean diet (e.g., accompanying meals). The goal of the present study was to test exercise and alcohol intake consumption in the context of real-life conditions.

Comment 6: The authors should use the future research aspect to explain a more rigorous test of the main hypotheses.

Response 6: Thank you for this comment. We have added this aspect to our discussion section [Ln 394-399].

Round 2

Reviewer 1 Report

Comments and Suggestions for Authors Thank you for applying the changes. The manuscript is well written, but describes statistically weak effects. Moreover, these effects may be the result of other factors than the authors think.